



# Technical Note: Hybrid Machine Learning Model for Bias Correction of UTLS Relative Humidity against IAGOS Observations in ERA5 Reanalysis

Mathieu Antonopoulos[1,2], Jérémie Juvin-Quarroz[1], and Olivier Boucher[1]

[1]Institut Pierre-Simon Laplace, Sorbonne Université/CNRS, Paris, France
[2]Institut Polytechnique de Paris, 91120 Palaiseau, France

**Correspondence:** Jérémie Juvin-Quarroz (jeremie.juvin-quarroz@ipsl.fr)

**Abstract.** Persistent contrail cirrus form in Ice-Supersaturated Regions (ISSRs) and are responsible for a large portion of aviation's non-$CO_2$ climate impact. Avoiding ISSRs through strategic flight rerouting has been proposed as a short-term mitigation strategy. However, accurate forecast of ISSRs is hindered by the difficulty of predicting Relative Humidity with respect to Ice, $RH_i$, at cruising altitude. Observations are problematic: Satellite-based global measurements carry large uncertainties while aircraft *in-situ* measurements offer a limited spatial coverage. On the contrary, ERA5 reanalysis offer a global estimate of $RH_i$, but it is known to exhibit a dry bias near the tropopause where ISSRs are located as well as significant random errors.

In this study, we develop a hybrid ensemble machine learning (ML) model to improve $RH_i$ estimates in the Upper Troposphere (UT) and Lower Stratosphere (LS) using ERA5 and aircraft measurements from the In-service Aircraft for a Global Observing System (IAGOS). The model combines a XGBoost regressor for drier conditions ($RH_i < 85\%$) and an Artificial Neural Network (ANN) for more humid cases ($RH_i > 85\%$). This hybrid approach significantly outperforms raw ERA5 data, reducing the mean absolute error from $13.7\%$ to $11.4\%$ and improving the Equitable Threat Score (ETS) for ISSR detection from $0.36$ to $0.44$. The greatest improvement is observed in the lower stratosphere, where the ETS increases by $0.18$ and the Mean Absolute Error (MAE) drops from $13.19\%$ to $10.71\%$. These improvements mark a key step toward more reliable identification of ISSRs, helping reduce the uncertainties that currently limit effective flight-rerouting strategies.

## 1 Introduction

Aviation contributes to climate change through both carbon dioxide ($CO_2$) emission and non-$CO_2$ effects. The latter may have accounted for 66% of aviation's total net Effective Radiative Forcing (ERF) in 2018 (Lee et al., 2021). The largest contributor to non-$CO_2$ ERF is the radiative forcing of condensation trails — contrails — which are linear-shaped ice clouds formed behind aircraft. The physical processes leading to contrail formation are well understood (Schmidt, 1941; Appleman, 1953; Schumann, 1996). While most contrails dissipate quickly and have a negligible climatic impact, those that form within Ice Supersaturated Regions (ISSRs) can persist for hours (Kärcher, 2018; Kärcher and Corcos, 2025), eventually developing into contrail cirrus clouds with microphysical and optical properties similar to those of natural cirrus (Li et al., 2023; Wang et al., 2024).



ISSRs are regions of the atmosphere where the relative humidity with respect to ice is supersaturated, $RH_i > 100\%$, which is a meta-stable thermodynamic state. The size of ISSRs has been estimated based on the distance flown within such regions by MOZAIC aircraft, showing that they can extend over $\mathcal{O}(10-100)$ kilometers (Gierens and Spichtinger, 2000). ISSRs are typically located at the tropopause level and are rather frequent across the globe (Gierens et al., 1999; Spichtinger et al., 2003a). Their formation is facilitated by the presence of upward winds and the meeting of cold and warm air streams (Gierens and Brinkop, 2012). The dynamics of ISSRs also present complex pattern as it has been determined that ISSRs' displacement may be slower than the surrounding wind field's (Hofer and Gierens, 2025).

On average, it is estimated that aircraft are spending 13.5% of their flight time in ISSRs (Gierens et al., 1999) and that only a small proportion of flights might be responsible for the majority of persistent contrail formation. For instance, it has been observed that only 2.2% of total flights are responsible for 80% of the total contrails ERF in the Japanese airspace (Teoh et al., 2020). These findings have motivated flight rerouting as a short-term mitigation strategy, *e.g.* Rosenow et al. (2018). Being able to meteorologically forecast ISSRs with enough accuracy would allow a slight deviation of the flight route to avoid the formation of persistent contrails, at the cost of a small amount of additional $CO_2$ emissions.

However, detecting ISSRs remains challenging, particularly through satellite observations (Gierens et al., 2004; Gettelman et al., 2006; Lamquin et al., 2012; Hegglin et al., 2013) due to their relatively narrow vertical extent (Spichtinger et al., 2003b; Wolf et al., 2023). The tropopause region, where ISSRs typically occur, also features steep vertical gradients in humidity (Stenke et al., 2008; Zahn et al., 2014), necessitating high vertical resolution for accurate characterization. Both remote measurements with lidar (Groß et al., 2014; Krüger et al., 2022) and in-situ observations, such as those made with balloon-borne instruments (Heymsfield et al., 1998) and with on-flight sensors (Diao et al., 2015b; Petzold et al., 2015), provide accurate measurements of $RH_i$, but are spatially and temporally limited.

Hence we must rely on the ERA5 reanalysis data (Hersbach et al., 2020) to obtain $RH_i$ values across the global atmosphere. However, it is known to suffer from a dry bias in the Upper Troposphere (UT) and Lower Stratosphere (LS) (Rolf et al., 2023). This bias, as well as significant random errors, can lead to considerable errors in $RH_i$ estimates, particularly in regions where contrails are likely to form. Previous work on the correction of humidity bias in ERA5 have shown potential for improvement of the ISSR detection in the UTLS. Wolf et al. (2025) applied a bivariate quantile mapping correction to ERA5 temperature and relative humidity fields, significantly reducing the dry bias. They report a reduction of the RHi bias from approximately -4.5% to -1.2% on average at pressure levels of 250, 225, and 200 hPa. This bias reduction leads to an improved classification of different contrail categories. Wang et al. (2025) proposed a machine learning (ML) approach to improve ERA5's RHi estimates in the UT using an Artificial Neural Network (ANN).

In this study, we present a hybrid ensemble machine learning approach to correct the dry bias in ERA5's $RH_i$ in the UTLS. We combine a XGBoost regressor (Chen and Guestrin, 2016) and an ANN, leveraging the complementary strengths of both models. ANNs are well-suited for modeling complex, non-linear relationships in multivariate data (Mihajlović and Nikolić, 2009), while tree-based models like XGBoost excel in handling structured data and often deliver state-of-the-art performance (Chen and Guestrin, 2016). We dynamically chose the model based on the initial $RH_i$ value provided by ERA5: the XGBoost is used for prediction in drier conditions ($RH_i < 85\%$), while the ANN is used for more humid cases ($RH_i > 85\%$). ISSR



formation involves more complex and non-linear atmospheric dynamics (Diao et al., 2014, 2015a), which are better captured
by ANNs due to their flexibility and non-linear representation capabilities. Effectively, the ANN captures subtle variations
relevant to ISSR classification, while the XGBoost model is used to correct bias under simpler thermodynamic conditions.

This paper is organized as follows. In Section 2 we describe the IAGOS and ERA5 datasets, the collocation procedure, filter-
ing, and feature engineering. Section 3 details the ANN and XGBoost architectures, training procedures, and the hybrid model.
Section 4 presents our results, showing improvements in regression and classification scores under different meteorological
conditions.

## 2  Data

In this study, we will use two complementary datasets. The ERA5 reanalysis data from the European Centre for Medium-Range
Weather Forecasts (ECMWF, Copernicus Climate Change Service, 2025), and the *in-situ* observations from the In-service
Aircraft for a Global Observing System (IAGOS, Petzold et al., 2015) program. ERA5 provides a global and consistent
estimate of atmospheric conditions through a wide range of variables, while IAGOS offers direct measurements of $RH_i$ at
cruise altitudes. Combining these datasets enables a supervised learning approach where the ERA5 features serve as model
inputs, while the $RH_i$ measurements from IAGOS serve as the target variable. The following subsections describe each dataset,
along with our preprocessing and feature engineering methodology.

We focus on the North Atlantic region, spanning latitudes from $40°N$ to $70°N$ and longitudes from $65°W$ to $5°E$ (Fig. 1).
This region shows a relatively uniform distribution of the IAGOS data and is known for frequent contrail formation due to high
air traffic and favorable atmospheric conditions. We selected the year 2022 as our study period as it offers recent atmospheric
conditions and comes after the maximum COVID disruption of international flights in 2020 and 2021. Our data processing
is implemented based on methodology from Wang et al., 2025, with some modifications to ensure proper separation of the
training and test datasets.

### 2.1  ECMWF reanalysis data (ERA5)

ERA5 data are provided on a regular latitude–longitude grid with a horizontal resolution of $0.25° × 0.25°$, corresponding to
an approximate grid spacing of 19.5 km at $45°N$ latitude. Vertically, the dataset includes 37 pressure levels extending from the
surface to the LS. We focus on pressure levels between 500 hPa and 125 hPa, specifically use the following levels: 500, 450,
400, 350, 300, 250, 225, 200, 175, 150, and 125 hPa. The extracted ERA5 variables provide a comprehensive description of
the atmospheric conditions (see Table 1 for a summary of all extracted variables and their units). The ERA5 data serve as input
for both statistical validation and the training of machine learning models. We use all variables as features for the model, and
the `RHi_ERA5` measurement as a baseline to evaluate model performance.



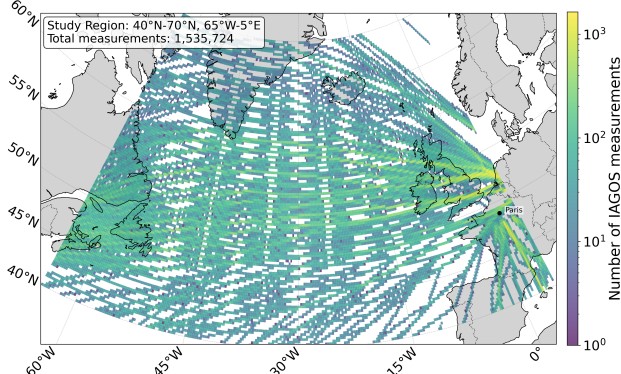

**Figure 1.** Density map of IAGOS data points in the North Atlantic region for the year 2022. The color scale indicates the number of data points per $0.25° \times 0.25°$ gridbox.

## 2.2 In-service Aircraft for a Global Observing System (IAGOS)

IAGOS is a European research initiative providing long-term, high-resolution *in-situ* atmospheric measurements to support
the understanding of atmospheric composition, air quality, and climate processes. Data are collected by equipping commercial aircraft, mostly Airbus A330s, with the 'P1 package' set of scientific instruments. Currently, the IAGOS fleet comprises ten aircraft operating worldwide. The $RH_i$ is derived from measured temperature, pressure, and water vapor mixing ratio, using the saturation vapor pressure formula by Sonntag (1994). The measurements are recorded at a temporal resolution of 4 seconds, corresponding to a spatial resolution of approximately 1 km at cruise speeds. However the data points are averaged over a
longer period because of the time lags involved with the instrument (Borella et al., 2024).

The spatial distribution of IAGOS measurements is inherently non-uniform as commercial flights tend to follow similar routes (cf. Fig. 1). Additionally, the limited number of aircraft in the IAGOS fleet contributes to a concentration of the data in some regions. The highest data density is found along transatlantic flight corridors and Western Europe. The North Atlantic and European regions are the most represented, as most flights from IAGOS are issued by European companies, and transatlantic
flights represent an important share of global aviation traffic. This area is of particular interest for contrail studies, given the frequent occurrence of ISSRs (Reutter et al., 2020). Most measurements are collected between 400 and 200 hPa, matching the standard cruise altitude.

To ensure data quality, we retain only samples flagged as 'good' by the IAGOS quality control system. We also restrict the dataset to measurements between 400 and 200 hPa. According to Rolf et al. (2023) and their analysis of the airborne
measurements, the P1 sensors used in IAGOS show limited accuracy in dry conditions ($RH_i < 10\%$). Hence, we exclude those samples from our dataset. After applying all these filtering steps for the year 2022 on the North-Atlantic region, we obtain a dataset comprising 1,535,724 data points from 678 flights.





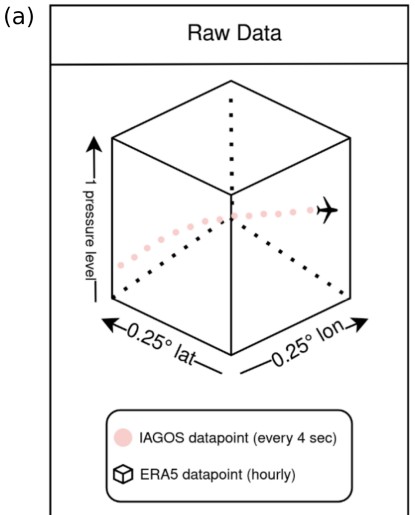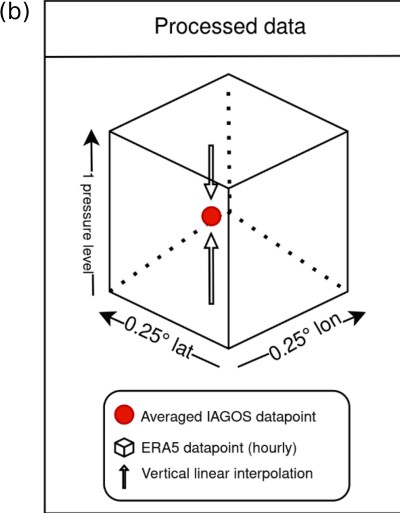

**Figure 2.** Illustration of the collocation process between IAGOS and ERA5 datasets. Diagram (a) presentes the datasets before collocation, while diagram (b) illustrates the data after preprocessing. IAGOS measurements are averaged within each ERA5 grid box, and the ERA5 variables are interpolated to match the mean altitude of the IAGOS point. The RH$_i$ is then recomputed based on the interpolated variables.

### 2.3 Data collocation

The IAGOS and ERA5 data have different resolutions in both time and space, hence we must proceed to a collocation of the

data. Once we collected and filtered the IAGOS data for each flight, we determine the closest point in ERA5 by matching the closest pressure level, longitude and latitude points (at a $0.25°$ resolution) and timestep (closest hour). Once we get these indices, we find the closest grid box in ERA5 to each IAGOS data point. The IAGOS points matching the same grid box are grouped. Given the four-second resolution of the data and the average grid box length of 19 km, each group contains up to 23 IAGOS points and 15 IAGOS points on average. Once grouped, the IAGOS data are averaged within one grid box to obtain

a single IAGOS point per ERA5 grid box and timestep. Given that points from same flights are usually located at a similar altitude within a grid box, we effectively horizontally average the IAGOS data for longitudes and latitudes. We further proceed to a vertical linear interpolation of the ERA5 data to match the averaged altitude of the IAGOS point. A visual representation of this process is given in Fig. 2. In order to maintain physical relationships between variables, we recalculate `RHi_ERA5` with the interpolated variables $T$ and $q$, which are respectively the air temperature and the specific humidity at the relevant pressure

value $p$. This is done using the Clausius-Clapeyron equation, with a Magnus form approximation for saturation vapor pressure over ice as descried by Alduchov and Eskridge (1996).

    The dataset is then separated into training, validation and test datasets. Careful separation is essential, as atmospheric conditions can exhibit strong spatial and temporal correlations. Without proper splitting, the model might train on samples nearly identical to those in the test dataset, potentially leading to overfitting and artificially high performance on validation and test





datasets. To avoid this issue while keeping a representative distribution across seasons, days, and hours, we implemented a by-day sampling strategy. Specifically, we sample five days for training, one day gap, one day for validation, one day gap, five days for training, one day gap, one day for testing and one day gap, uniformly throughout the year. After collocation and split, we obtain $64,712$ training samples, $7,025$ validation samples and $6,085$ test samples. To address the underrepresentation of high $RH_i$ values and reduce potential bias, data augmentation is applied to the training dataset: samples with $RH_i$ above $120\%$

are oversampled by a factor of three, while those below $20\%$ $RH_i$ are undersampled by a factor of $0.4$. This approach results in a $RH_i$ distribution that is more suitable for our needs.

Finally, we ensure that our processed data are representative of ISSRs characteristics by comparing it to the results in Gierens and Brinkop (2012). We report the distribution of vorticity and divergence in Fig. A1.

## 2.4 Feature engineering

ERA5 provides an extensive set of variables, but to further improve the input data and capture new relation (Fig. 3), additional derived variables are added as input features. Following the methodology proposed by Wang et al. (2025), we add for each data point, the selected variables two pressure levels above and below the current pressure level (totaling five levels) as well as data from two hours and six hours prior to the current timestep. In addition we extracted several features from the original ERA5 and IAGOS data to enrich the model inputs and improve the correction of the $RH_i$ estimates. These features aim to capture the

vertical structure and dynamics of the atmosphere.

To capture the vertical temperature profile, we compute temperature gradients at different scales. These include the gradient between the current level and the two levels immediately above (`T_grad_up`), between the current level and the two levels immediately below (`T_grad_down`), a centered gradient between one level above and one below (`T_grad_centered`), and a lower-resolution gradient between two levels above and two below (`T_grad_overall`), in K hPa$^{-1}$. Similarly, gradients of

145 the vorticity (`vo_grad_up`, `vo_grad_down`, `vo_grad_centered`, `vo_grad_overall`), in s$^{-1}$ hPa$^{-1}$, and the $RH_i$ (`RHi_grad_up`, `RHi_grad_down`, `RHi_grad_centered`, `RHi_grad_overall`), in $\%$, are computed following the same pattern.

To account for the diurnal and seasonal variability in atmospheric conditions, we embedded time information using cosine functions. The local time at the observation point was computed from the UTC hour and longitude, then encoded using sine

and cosine transforms to capture daily cycles (`cos_hour`, `sin_hour`). Similarly, the day of the year (`cos_day`, `sin_day`) is encoded to reflect seasonal effects. These four quantity are normalized by $2\pi$.

These engineered features were incorporated into the training data to allow the model to better represent the complex atmospheric processes influencing the $RH_i$ (cf. Table 1). Some of the engineered features proved to be statistically meaningful. In particular, the temperature gradient showed a stronger correlation with $RH_i$ than the original temperature variable. Time

embeddings exhibited almost null direct correlation with RHi. However, they were identified as important contributors in the feature importance analysis of the ANN, indicating that their effect may be captured in more complex, non-linear relationships within the model (cf. Fig. 3 and Table 4).



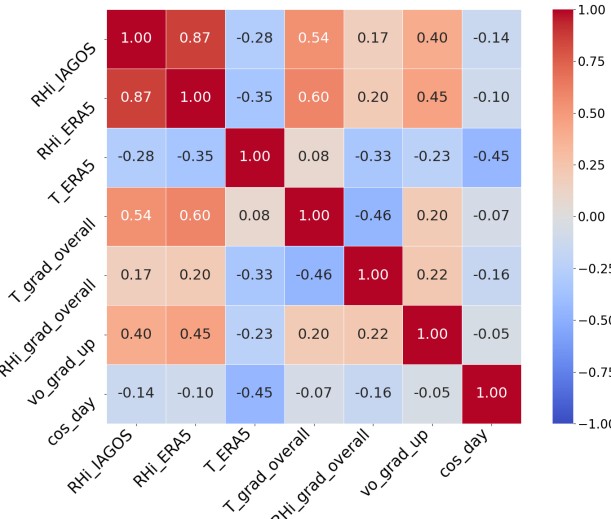

**Figure 3.** Correlation matrix showing the relationships between selected engineered features, the target variable RHi_IAGOS, and the initial ERA5 estimate. The engineered gradient features show a relatively strong correlations with the target RHi_IAGOS. The time embeddings (cos_day) exhibit low direct correlation with RHi_IAGOS.

Data is then categorized according to distinct meteorological conditions following methodology from Wang et al. (2025). To determine the cloud presence, we used the cloud ice water content (ciwc_ERA5) variable. If ciwc equals zero at the current pressure level and two levels above and below it ($\pm 2$ pressure levels), the data point is classified as clear sky. If any of these five levels has a non-zero ciwc value, the point is classified as cloudy. Additionally, we seperated the points from the UT and LS. Commercial flights cruise at an atmospheric level called the tropopause, which is the interface between the troposphere and the stratosphere. This separation helps to account for the differing thermodynamic properties and humidity distributions between the troposphere and stratosphere. The classification is based on potential vorticity (pv_ERA5). Data points are labeled as UT if potential vorticity is less than 2 PVU (Potential Vorticity Units). Conversely, points with PV > 2 PVU are classified as LS. We use the typical conversion rate of $1\ \mathrm{PVU} = 10^6\ \mathrm{K\ kg^{-1}\ m^2\ s^{-1}}$. In the test dataset, 2641 points are classified as Cloudy-Sky, 3444 as Clear-Sky, 4566 as LS and 1519 as UT.

## 2.5 Evaluation Metrics

The performance of the model is assessed using different metrics. Since we primarily solve a regression task, we use the MAE, expressed as $\mathrm{RH_i}$ percentage points. However, MAE alone does not fully capture the model's ability to correct the biases in ERA5. The aim of the model is to accurately identify ISSRs, which requires a classification perspective. Therefore, following common practices in ISSR prediction (such as in Wang et al., 2025 and Wolf et al., 2025), the Equitable Threat Score (ETS) is used as a complementary evaluation metric. The ETS is a skill score that quantifies the accuracy of a binary classification (e.g.,



| Category | Feature Name | Description | Unit |
|---|---|---|---|
| ERA5 variables | T_ERA5 | Air temperature | K |
| | q_ERA5 | Specific humidity | $kg \cdot kg^{-1}$ |
| | w_ERA5 | Vertical wind speed | $Pa \cdot s^{-1}$ |
| | u_ERA5 | Zonal (west-east) wind component | $m \cdot s^{-1}$ |
| | v_ERA5 | Meridional (south-north) wind component | $m \cdot s^{-1}$ |
| | pv_ERA5 | Potential vorticity | $K \cdot m^2 \cdot kg^{-1} \cdot s^{-1}$ |
| | ciwc_ERA5 | Cloud ice water content | $kg \cdot kg^{-1}$ |
| | geopt_ERA5 | Geopotential | $m^2 \cdot s^{-2}$ |
| | vo_ERA5 | Relative Vorticity | $s^{-1}$ |
| | d_ERA5 | Horizontal divergence | $s^{-1}$ |
| | RHi_ERA5* | $RH_i$ | % |
| Engineered features | T_grad_up | Temp. gradient: current/upper level | $K \cdot hPa^{-1}$ |
| | T_grad_down | Temp. gradient: current/lower level | $K \cdot hPa^{-1}$ |
| | T_grad_centered | Temp. gradient: $\pm 25$ hPa | $K \cdot hPa^{-1}$ |
| | T_grad_overall | Temp. gradient: $\pm 50$ hPa | $K \cdot hPa^{-1}$ |
| | RHi_grad_up | $RH_i$ gradient: current/upper level | $\% \, hPa^{-1}$ |
| | RHi_grad_down | $RH_i$ gradient: current/lower level | $\% \, hPa^{-1}$ |
| | RHi_grad_centered | $RH_i$ gradient: $\pm 25$ hPa | $\% \, hPa^{-1}$ |
| | RHi_grad_overall | $RH_i$ gradient: $\pm 50$ hPa | $\% \, hPa^{-1}$ |
| | vo_grad_up | Vorticity gradient: current/upper level | $s^{-1} \cdot hPa^{-1}$ |
| | vo_grad_down | Vorticity gradient: current/lower level | $s^{-1} \cdot hPa^{-1}$ |
| | vo_grad_centered | Vorticity gradient: $\pm 25$ hPa | $s^{-1} \cdot hPa^{-1}$ |
| | vo_grad_overall | Vorticity gradient: $\pm 50$ hPa | $s^{-1} \cdot hPa^{-1}$ |
| | PVU | Potential vorticity in PVU | PVU |
| | cloudy | Ice-cloud flag | $\{0, 1\}$ |
| | cos_hour | Cosine of local time | — |
| | sin_hour | Sine of local time | — |
| | cos_day | Cosine of day in year | — |
| | sin_day | Sine of day in year | — |
| IAGOS variables | p_IAGOS | IAGOS pressure | hPa |
| | **RHi_IAGOS** | IAGOS $RH_i$ | % |

**Table 1.** Summary of all features used for training, grouped by category. ERA5 variables include values at current pressure level and context, $\pm 2$ pressure levels, and 2 h and 6 h prior current timestep (i.e. 7 versions for each ERA5 variable). The target IAGOS value is indicated in **bold**. RHi_ERA5 is recomputed after linear interpolation to maintain consistency and physical dependencies

detecting ISSRs where $RH_i >= 100\%$) while accounting for hits that could occur by random chance. The ETS is defined as

$$ETS = \frac{TP - r}{TP + FP + FN - r},$$ (1)

where *TP* denotes true positives (correctly predicted ISSRs), *FP* denotes false positives (non-ISSRs incorrectly predicted as ISSRs), *FN* denotes false negatives (ISSRs missed by the model), *TN* denotes true negatives (correctly predicted non-ISSRs), and *r* is the number of hits expected by random chance, computed as

$$r = \frac{(TP + FP)(TP + FN)}{TP + FP + FN + TN},$$ (2)





The ETS adjusts for random hits and is particularly suited for evaluating rare event prediction. It ranges from $-1/3$ (worse than random chance) to $1$ (perfect prediction), with $0$ indicating no better results than random guessing. Our baseline for evaluating model improvement is the ERA5 $RH_i$ prediction, which shows a $0.36$ ETS and $13.7\%$ MAE.

## 3 Hybrid Machine Learning Model development

Correcting the dry bias in ERA5's $RH_i$ estimates in the UTLS presents a significant challenge due to the inherently non-
185 linear nature of atmospheric processes. To address this, we propose a hybrid machine learning approach that combines two complementary models: an ANN and a XGBoost. The ANN is well-suited for capturing complex, non-linear dependencies, particularly under high humidity conditions, while XGBoost performs robustly with structured input data, especially in drier regimes.

  The hybrid strategy dynamically selects the appropriate model for prediction based on the ERA5-estimated $RH_i$ value.
Specifically, for samples where ERA5 RHi is below $85\%$, predictions are taken from the XGBoost model, which has demonstrated superior performance in lower-humidity environments. Conversely, for $RH_i$ values exceeding $85\%$, the ANN model is employed, leveraging its capacity to extract highly non-linear patterns, such as those associated with ISSRs. The $85\%$ threshold was chosen based on two key considerations. First, ERA5 is known to exhibit a MAE of approximately $15\%$ in humid conditions. By subtracting this from the $100\%$ threshold typically used to identify ISSRs, we ensure that the ANN model
is prioritized in critical high-humidity cases. Second, validation dataset performance confirmed that this hybrid architecture consistently outperforms either model used in isolation.

  Firstly, the ANN component is trained to predict $RH_i$ using the input features summarized in Table 1. Data are normalized using min–max scaling, with the RHi range extended to $200\%$ to reduce sensitivity to outliers and mitigate dry bias. The final network architecture includes three hidden layers with 100 neurons each, employing He initialization and ReLU acti-
200 vation functions. Batch normalization and dropout regularization are applied between layers to stabilize training and reduce overfitting. The output layer uses a linear activation function to predict $RH_i$ , denoted as `RHi_ANN`.

  Training is conducted using the Adam optimizer (Kingma and Ba, 2014) with a learning rate of $0.001$, decay rate of $5\cdot10^{-3}$, and momentum of $0.98$. The model is trained over 150 epochs with a batch size of 1024, and early stopping is employed to halt training if validation loss does not improve over 20 consecutive epochs. A custom loss function, based on the Mean Squared
Error (MSE), is used to emphasize accuracy in high-$RH_i$ conditions. The loss function is defined as:

$$\mathcal{L}(y,\hat{y}) = \frac{1}{n}\sum_{i=1}^{n} w_i \cdot (y_i - \hat{y}_i)^2, \qquad \text{with} \quad w_i = 1 + y_i^\alpha. \tag{3}$$

  In Eq. 3, $y_i$ is the target value at index $i$, $\hat{y}_i$ is the predicted value at index $i$, $n$ is the number of samples, and $\alpha$ is a weight factor. The weight parameter $(\alpha)$ that defines the weight of the $RH_i$ value is denoted `WEIGHT_FACTOR`. This weighted approach prioritizes performance in ISSRs, addressing the ERA5 dry bias. Hyperparameters were optimized via grid search on
the train dataset as detailed in Table 2.





| Hyperparameter | Values |
|---|---|
| BATCH_SIZE | [512, **1024**, 2048] |
| DROPOUT_RATE | [0.0, **0.1**, 0.3] |
| EPOCHS | [100, **150**] |
| HIDDEN_DIM | [64, **100**, 128, 256, 512] |
| LEARNING_RATE | [0.1, 0.01, **0.001**, 0.0001] |
| NUM_LAYERS | [2, **3**, 4, 5] |
| WEIGHT_DECAY | [**0.005**, 0.0001, 0.00001] |
| MOMENTUM | [0.85, 0.95, **0.98**, 0.99] |
| PATIENCE | [10, **20**, 30] |
| MIN_DELTA | [**0.0001**] |
| WEIGHT_FACTOR | [3, 10, **30**] |
| OPTIMIZER | [**adam**, sgd] |
| SCHEDULER_FACTOR | [**0.5**] |
| SCHEDULER_PATIENCE | [10, **20**] |

**Table 2.** Hyperparameter grid used for tuning the ANN model. Each row lists the set of values tested for a specific hyperparameter during the grid search process. The values in **bold** indicate the configuration that achieved the best performance on the validation dataset and was selected for final training.

Secondly, the XGBoost model uses the same input feature set as the ANN, including engineered variables. XGBoost is an efficient implementation of gradient boosting known for its scalability, accuracy, and interpretability. The final model was trained with $100$ estimators, a learning rate of $0.1$, a maximum tree depth of $4$, a subsampling ratio of $0.9$, and a column sampling ratio per tree of $0.8$. In Table A1, we report the performance comparison for a set of ML models.

## 4 Results

The hybrid model achieved an ETS of $0.44$ and a MAE of $11.37\%$. Compared to ERA5, it showed an improvement of $+0.08$ in ETS and $-2.33\%$ in MAE. Fig. 4 compares scatter plots of our different models predictions on the test dataset to the baseline ERA5 estimates, see Fig. 4(a). Fig. 4(c) shows that the XGBoost model was able to correct low $RH_i$ values with great accuracy, consequently showing strong MAE improvement. However, it struggles with higher $RH_i$ values, resulting in poor ETS. The ANN, Fig. 4(b), shows a strong ETS, indicating its ability to classify ISSRs, but it overestimates the lower $RH_i$ values leading to a higher MAE. The hybrid model shown in Fig. 4(d) combines the strengths of both models, achieving a good balance between ETS and MAE. It effectively corrects low $RH_i$ values while maintaining a strong improvement in the ETS.

The $RH_i$ estimates are effectively improved by combining the strengths of both models, leveraging the ANN's ability to capture non-linear relationships and the XGBoost's robustness in handling drier conditions. The hybrid model achieves a higher ETS comparable to the ANN's while maintaining a lower MAE than both models. This hybrid approach demonstrates the potential of combining different machine learning techniques to improve predictions in complex atmospheric conditions. Table 3 summarizes the performance of all models in terms of ETS and MAE.





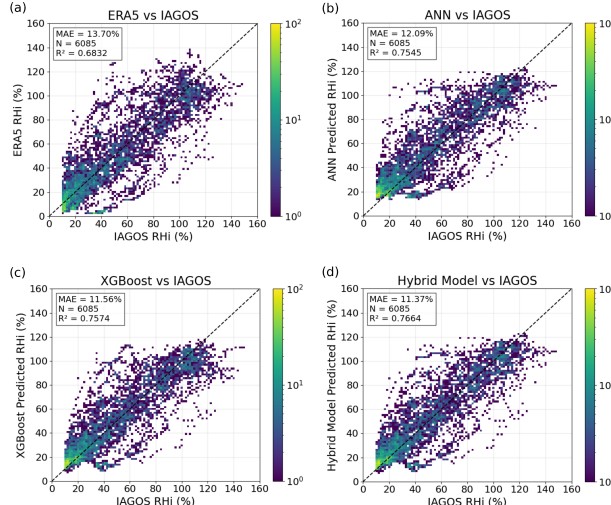

**Figure 4.** Comparison of RHi predictions between different models with respect to the target value (`RHi_IAGOS`). (a) presents the baseline ERA5 reanalysis, (b) ANN model, (c) XGBoost model, and (d) the hybrid ensemble model. All predictions are made on the test dataset between 400 and 200 hPa over the North Atlantic region (cf. Fig. 1) for the year 2022.

| Model | ETS | MAE (%) | Description |
|---|---|---|---|
| ERA5 (Baseline) | 0.36 | 13.70 | Original reanalysis data (baseline) |
| ANN | 0.44 (+0.08) | 12.09 (-1.61) | Strong ISSR classification, moderate error reduction |
| XGBoost | 0.35 (-0.01) | 11.56 (-2.14) | Weak ISSR classification, strong error reduction |
| Hybrid Model | **0.44** (+0.08) | **11.37** (-2.33) | Strong ISSR classification, strong error reduction |

**Table 3.** Performance comparison of models for $RH_i$ correction. Improvement relative to ERA5 baseline shown in parentheses.

Table 4 presents the most important features used by the ANN and XGBoost models. As expected we find `RHi_ERA5` as the most important feature. This table shows that the models relied on different features for predictions: the XGBoost model relies more heavily on features with stronger linear relationships to the target, such as the vertical temperature gradients, while the ANN is able to exploit more complex, non-linear patterns, like the time embeddings. This observation highlights the complementary nature of the two models, with the ANN capturing complex relationships (often associated with more humid regions) and the XGBoost focusing on more linear patterns.

The performance of the hybrid model relative to the baseline ERA5 are assessed for both UT and LS under clear- and cloudy-sky conditions. Results are shown in Fig. 5. The best improvement in MAE is observable in the LS, from 13.19% to 10.71%. This is explained because the model performs a better correction on lower $RH_i$ values and the LS counts significantly less ISSRs (see Fig. 6). In Table 5, we report the predictions performance for each of the cases.





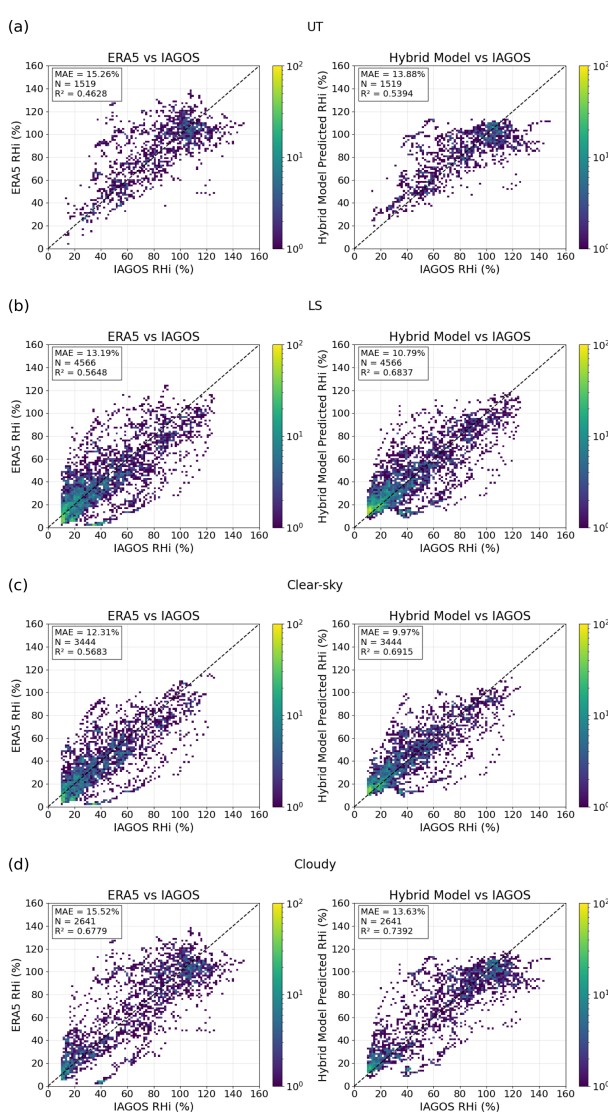

**Figure 5.** Comparison of ISSR prediction performance between ERA5 and the hybrid model across four meteorological conditions. These are the predictions made on the test dataset between 400 and 200 hPa over the North Atlantic region (cf. Fig. 1) for the year 2022.




| ANN Model | | XGBoost Model | |
|---|---|---|---|
| Feature | Importance | Feature | Importance |
| RHi_ERA5 | 0.14 | RHi_ERA5_prior_2h | 0.31 |
| cos_day | 0.09 | RHi_ERA5 | 0.13 |
| RHi_ERA5_up1 | 0.06 | Cloudy | 0.12 |
| cos_hour | 0.05 | T_grad_down | 0.09 |
| Cloudy | 0.04 | RHi_ERA5_up1 | 0.06 |

**Table 4.** Top five most important features for the ANN and XGBoost models. As expected, RHi_ERA5 ranks highest in both models due to its strong correlation with the target variable. Several engineered features demonstrate strong importance, specifically the temperature gradient, the binary cloud presence indicator, and the time embeddings. Time embedding features show strong importance for the ANN model but not for XGBoost, indicating that each model captures different aspects of the underlying atmospheric patterns.

| Condition | Model | TP | FN | FP | TN | ETS |
|---|---|---|---|---|---|---|
| UT | ERA5 | 24.09 | 11.45 | 13.50 | 50.95 | 0.30 |
| | HYB | 24.36 | 11.19 | 10.66 | 53.79 | 0.35 (+0.05) |
| LS | ERA5 | 1.18 | 3.29 | 1.34 | 94.20 | 0.19 |
| | HYB | 2.01 | 2.45 | 0.72 | 94.81 | 0.37 (+0.18) |
| Cloudy | ERA5 | 14.96 | 8.56 | 9.66 | 66.83 | 0.33 |
| | HYB | 16.02 | 7.50 | 7.08 | 69.41 | 0.42 (+0.09) |
| Clear-sky | ERA5 | 0.73 | 2.85 | 0.32 | 96.11 | 0.18 |
| | HYB | 1.13 | 2.44 | 0.23 | 96.20 | 0.29 (+0.11) |

**Table 5.** Comparison of ERA5 and hybrid model (HYB) prediction performance against IAGOS observations across different meteorological conditions on the test dataset. We show the percentage of True Positive (TP), False Negative (FN), False Positive (FP) and True Negative(TN) per condition.

Finally, Fig. 6 presents the number of correctly predicted ISSRs for the various cases. We observe that the hybrid model correctly predicts significantly more ISSRs than ERA5 in the LS, 92 out of 204, compared to 54 out of 204 for ERA5. Overall, the hybrid model improves ISSR detection by approximately 10% across all conditions. In the UT, the improvement is less significant, with only 4 additional ISSRs correctly classified.




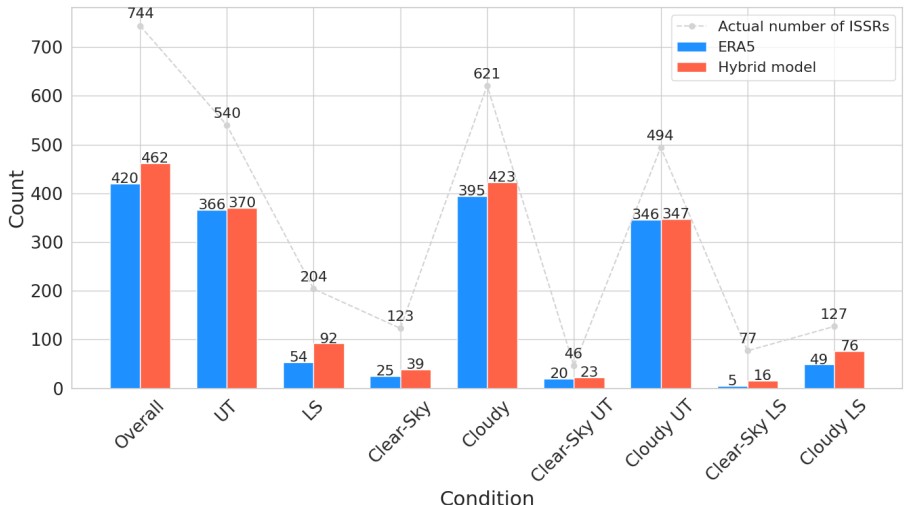

**Figure 6.** Number of correctly predicted ISSRs by ERA5 and the hybrid model compared to IAGOS ground truth across different meteorological conditions on the 2022 test dataset. Blue bars indicate true positives from ERA5, red bars from the hybrid model, and the dashed curve represents the total number of actual ISSRs in each condition based on IAGOS observations. The largest improvement is observed in the LS. The far left bars show the overall comparison across all conditions.

## 5 Conclusions

In this study, we used a hybrid ensemble model to improve $RH_i$ from ERA5 reanalysis in the UTLS using *in-situ* measurement from IAGOS. The dry bias of ERA5 is considerably reduced and the number of correctly predicted ISSRs is increased, especially in the LS where it nearly doubles.

The hybrid model is built on two complementary models: a XGBoost regressor that excels in dry conditions ($RH_i < 85\%$), and an ANN that performs better in more humid regions ($RH_i > 85\%$). The hybrid model demonstrated significant improvements compared to the initial ERA5 estimates, both in terms of MAE and ETS. It achieved an MAE of $11.37\%$, compared to $13.7\%$ for ERA5, and improved the ETS from $0.36$ to $0.44$. The best improvement was observed in the LS, where the hybrid model increased ETS by $+0.18$ and reduced MAE from $13.19\%$ to $10.71\%$. The hybrid model consistently outperformed ERA5 across all defined meteorological conditions, detecting approximately $10\%$ more ISSRs overall.

However, the model presents limitations, especially in predicting very high RHi values (above $120\%$), likely due to the very limited availability of such samples in the dataset. Although ISSR detection has improved, the reanalysis still falls short of the requirements to support effective contrail avoidance strategies. Future work could explore advanced downsampling techniques to improve collocation between IAGOS and ERA5 datasets despite their different spatial and temporal resolutions.



*Data availability.* The IAGOS data are available on the IAGOS Data Portal (Boulanger et al., 2019). The ERA5 reanalysis data are available from the Copernicus Climate Change Service (C3S) Climate Data Store (CDS) (Hersbach et al., 2023).

# Appendix A

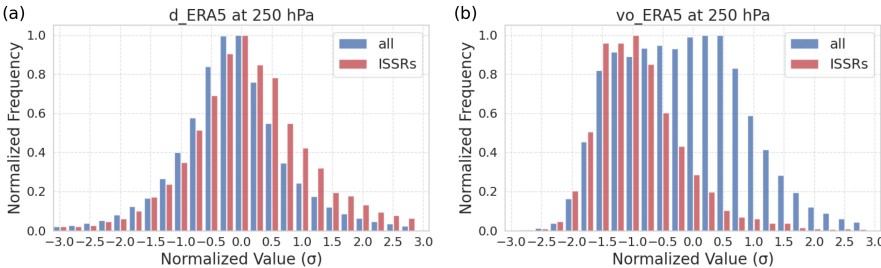

**Figure A1.** Histograms of vorticity (a) and divergence (b) for the North-Atlantic region in 2022 on the 250 hPa level. This pressure level corresponds to the UT, where ISSRs are frequently observed. The blue bars refer to all points region while the red ones refer to ISSRs only. The data are normalized using the standard deviation of all grid boxes. We compared these distributions with studies of dynamical characteristics of ISSRs from Gierens and Brinkop (2012). By ensuring that our processed data exhibit comparable statistical properties, we confirmed the reliability of our data processing pipeline.



| Model | Train $R^2$ | Test $R^2$ | Train MAE | Test MAE | Train MSE | Test MSE | Train ETS | Test ETS |
|---|---|---|---|---|---|---|---|---|
| Decision Tree | 0.87 | 0.66 | 8.76 | 13.64 | 12.79 | 19.45 | 0.47 | **0.40** |
| Random Forest | **0.99** | 0.74 | **1.37** | 12.12 | **2.54** | 17.00 | **0.92** | 0.29 |
| Gradient Boosting | 0.87 | 0.75 | 9.28 | 11.84 | 12.90 | 16.56 | 0.46 | 0.31 |
| Hist GB | 0.89 | 0.74 | 7.36 | 11.76 | 11.73 | 17.00 | 0.58 | 0.34 |
| AdaBoost | 0.69 | 0.61 | 17.0 | 17.79 | 19.88 | 20.84 | 0.06 | 0.02 |
| XGBoost | 0.87 | **0.76** | 9.36 | 11.56 | 13.05 | **16.25** | 0.48 | 0.35 |
| LightGBM | 0.87 | 0.76 | 9.19 | **11.55** | 12.82 | 16.30 | 0.48 | 0.32 |
| Linear Regression | 0.77 | 0.73 | 12.30 | 12.74 | 16.98 | 17.41 | 0.29 | 0.35 |
| Ridge | 0.77 | 0.73 | 12.25 | 12.62 | 16.94 | 17.34 | 0.29 | 0.37 |
| Lasso | 0.75 | 0.72 | 13.53 | 13.71 | 17.95 | 17.69 | 0.08 | 0.11 |
| ElasticNet | 0.75 | 0.72 | 13.13 | 13.38 | 17.77 | 17.56 | 0.12 | 0.14 |
| Bayesian Ridge | 0.77 | 0.73 | 12.25 | 12.62 | 16.95 | 17.37 | 0.29 | 0.37 |
| SGD Regressor | 0.77 | 0.72 | 12.23 | 12.62 | 17.07 | 17.41 | 0.32 | 0.39 |
| KNN | 0.99 | 0.60 | 1.76 | 15.18 | 3.54 | 21.09 | 0.88 | 0.32 |

**Table A1.** Model performance comparison across regression and classification metrics for a set of ML model available in Scikit-learn (Pedregosa et al., 2011). The train and test dataset are described in Section 2.3. Best values per metric are in **bold**.

*Author contributions.* OB and JJQ conceived and designed the experiments; MA conducted the experiments and analyzed the data; MA and
260 JJQ wrote the manuscript; OB provided critical review.

*Competing interests.* None of the author has any competiting interests.

*Acknowledgements.* The authors thank Ziming Wang for useful discussions on her work which has initiated our study.



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
