# Peer review of "Technical Note: Hybrid Machine Learning Model for Bias Correction of UTLS Relative Humidity against IAGOS Observations in ERA5 Reanalysis"

_EGUsphere, 2025_

## Author Comment (AC1)

**Response to Reviewer #1**

We would like to thank the anonymous Reviewer for their careful review. We have taken all of the comments into account, improving the accuracy and presentation of the results in our manuscript. Below is our response to the comments on a point-by-point basis. The questions and remarks from the Reviewer are recalled in italic.

**Overall**

*This is a nice technical note expanding on the work of Wang et al 2025. I think the manuscript deserves to be published after improving the clarity of the presentation and considering the major comments below. Ideally the manuscript would be accompanied by example training code in the author's language of choice.*

We thank the Reviewer for the overall positive assessment of our manuscript. We will attach to the article an example training code written in python.

**Major Comments**

*- L3 & L46: Many publications on this topic often make some form of the statement "[There are] considerable errors in RH_i estimates" which makes "accurate forecast of ISSRs [difficult]." I'm interested to see more analysis on the type and distribution of errors to better understand how ISSR forecast errors will result in ineffective (or inefficient) avoidance measures. In our experience, RH_i (and ISSRs) have high pointwise error, but overall ISSR regions are (generally) spatially and temporally correlated with ISSR forecasts.*

The Reviewer emphasises an essential distinction between correctly predicting RHi and correctly predicting ISSRs. It is true that errors on the (vertical and horizontal) location of ISSRs may be less than errors on the intensity of the supersaturation pointwise. In our study, we are taking care of the latter and, therefore, we are only addressing part of the problem. As a consequence, we weaken our statement and modify the line 3 as follows

> "However, accurate prediction of the Relative Humidity with respect to Ice, RHi, distribution within ISSRs at cruising altitude remains difficult."

*- L253: What are the requirements to support effective contrail avoidance strategies?*

Effective contrail avoidance strategies will require efficient, quick, and accurate meteorological predictions of ISSRs. However, the specific performance requirements for such predictions remain unclear. Several strategies have been proposed (avoiding ISSRs entirely or avoiding both ISSRs and their surrounding regions) each of which may impose different accuracy requirements on the forecasts. We qualify our statement and modify the line 253 as follows:

> "Although ISSR detection has improved, reanalysis products still exhibit limitations to support potential contrail avoidance strategies, especially in the absence of clearly established performance requirements."

*- L51: Wang et al 2025 published a ANN humidity correction methodology. This publication adds an XGBoost regression for RH_i < 85%, and a different training/validation data split. Given the*

*similarities, this line  deserves a whole paragraph describing the differences with Wang 2025, and how this methodology aims to improve on the previous work.*

Line 51: This study is building on the methodology by Wang et al. (2025). One of the key differences is the rigorous sampling strategy to strictly prevent temporal leakage between the training and testing/validation sets. This new splitting strategy revealed much weaker performance from the standalone ANN, especially in terms of MAE on dry values. To address this, we developed the hybrid architecture presented in this study. Thus, the main differences from the original work are the strict data splitting strategy, the introduction of new engineered features (detailed in Section 2.4), and the integration of an XGBoost model. Additionally, our study focuses on the North Atlantic region as it is a hot spot for air traffic and a potential region for contrail avoidance trials. Furthermore, it offers a better coverage by the IAGOS dataset.
We think that an additional paragraph highlighting the main differences with Wang et al. (2025) would be redundant. We note that additional point of comparison have been added:
- line 73: "Unlike Wang et al. (2025), we focus on the North Atlantic region, spanning latitudes from 40◦N to 70◦N and longitudes from 65◦W to 5◦E (Fig. 1)."
- line 75: "For these reasons, it is thought to be a potential region for contrail avoidance trials."
- line 79: "Our data processing is implemented based on methodology from Wang et al., 2025, with some modifications to ensure proper separation of the training and test datasets to avoid temporal leakage."
- line 126: "This differs from Wang et al. (2025) in two ways: i) we use five days for training (four days in Wang et al. (2025)), ii) an additional day gap is used after the validation or testing day to prevent correlations with the following training period."

*- L74: What kind of biases in the weather might this domain selection introduce? Have you tested how well your models apply outside this domain?*

Our study focuses on the North Atlantic, adapting the model to other regions of interest would require tuning both the model and the engineered features to the specific thermodynamic properties of the new region, as these differences can impact performance. However, this expansion presents a major challenge because we rely on the IAGOS dataset as ground truth, and measurements are sparse or non-existent in many other regions. This lack of data complicates the training process and severely limits our ability to robustly assess the actual performance of the model globally.

*L83: Did you consider model levels? It may be worth exploring if the higher vertical resolution would improve your results.*

We acknowledge that using ERA5 model levels could potentially benefit the results due to their higher vertical resolution. However, this approach was not implemented in the final analysis due to the large size of the dataset and the significant computational resources required for processing. We consider the current approach to offer good tradeoff between precision and computational performance, without compromising the main conclusions.

*- L117-121: How did you interpolate the values for T and q? Linear interpolation in q introduces bias when working with coarse pressure levels.*

We performed a linear interpolation in the vertical dimension for T and q to map the ERA5 pressure levels to the aircraft altitude. While we acknowledge that linear interpolation of humidity can introduce biases over large vertical distances, we found this method to be sufficient given the vertical resolution

of ERA5 in the UTLS region and the scope of our study. We experimented with polynomial interpolation but the resulting datapoints did not match the IAGOS observations as closely as the linear approach. Leveraging the higher vertical resolution from ERA5 model levels in future work could mitigate the bias introduced by interpolation.

*- Table 1: Teoh et al 2024 introduced a latitude correction for the humidity correction. Should latitude be a feature?*

During our data analysis, we explored a wide range of potential features, including latitude, to evaluate their impact on the model. However, our feature selection process revealed no significant statistical relationship between latitude and RHi in our dataset, and its inclusion did not lead to a meaningful improvement in predictive performance. Hence, latitude was filtered out at the feature selection process.

**Minor Comments**

*- L31: "are spending" -> "spend"*

We have corrected the typo.

*- L33: Suggest using stats from more recent Teoh, R. et al. (2024) "Global aviation contrail climate effects from 2019 to 2021," Atmospheric Chemistry and Physics, 24(10), pp. 6071–6093. Available at: https://doi.org/10.5194/acp-24-6071-2024.*

We have changed to a more recent publication and adapted the text accordingly:
"For instance, it has been estimated that only 2.7% of total flights are responsible for 80% of the total contrail RF in 2019 (Teoh et al., 2024)."

*- L37: Its worth motivating why we need to detect ISSRs. Its presumed that the reader knows "to meteorologically forecast ISSRs with enough accuracy" we need ISSR detections. May want to add context e.g. "Global ISSR forecasts are generally derived for numerical weather forecasting systems, or nowcast from in situ measurements or inferred from remote sensing. Both approaches rely on accurate detections of ISSRs, in the first case to validate models, or in the second through measurements"*

We have added context to motivate the ISSR detection by including your suggested sentence.

*- L44: Not just ERA5 - any numerical weather prediction system. I'd flip this around - numerical weather prediction models provide a comprehensive prediction across the global atmosphere. ERA5 is a highly trusted source of numerical weather prediction.*

We have modified the statement as follows:
"Hence we must rely on numerical weather prediction models to obtain RHi values across the global atmosphere. We use the ERA5 reanalysis data (Hersbach et al., 2020) as it is highly trusted source of numerical weather prediction."
In addition, we modify line 5 as follows:
"On the contrary, reanalyses offer a global estimate of RHi, but it is known to exhibit a dry bias near the tropopause where ISSRs are located as well as significant random errors."

*- L45: Define what a dry-bias means*

We have defined the dry bias as follows:
"However, it is known to suffer from a dry bias in the Upper Troposphere (UT) and Lower Stratosphere (LS) (Rolf et al., 2023), which means that the model often predict lower relative humidity values compared to the observed values."

*- L48: Other publications with humidity correction: (constant) Schumann, 2012; Schumann et al., 2015; Teoh et al., 2020; Schumann et al., 2021; (piecewise function) Teoh et al 2022; Teoh et al 2024; (quantile mapping) Platt et al 2024*

We have included additional references for humidity corrections (Teoh et al., 2022; Platt et al., 2024).

*- L55: This sentence sounds like an LLM. I'd move L59-L61 up front, remove this sentence, and then have L57-58. Can you be more specific as to why you chose the hybrid model? From this description it sounds like you used XGBoost for compute performance reasons rather than accuracy.*
We think that the sentence at line 55 is important as it describes the ANN and three-based models in a general manner, before focusing on the case of ISSRs.
Concerning the decision to use XGBoost for the hybrid model, it was driven by accuracy rather than computational performance. As discussed in section 4 and illustrated in Figure 4, XGBoost significantly outperforms the ANN in reducing the MAE in drier conditions. However, the ANN excels at capturing the non-linear dynamics associated with high humidity and achieves a much higher ETS (resulting in a better classification of ISSRs). Therefore, the hybrid approach was selected to leverage the complementary strengths of the models to achieve a better classification performance while maintaining a strong overall accuracy.

*- L94: How long is the "longer period"?*

We have specified "a longer period of few minutes"

*- L104: Just confirming that IAGOS accuracy is a function of RH_i or of absolute humidity. I had remembered that humidity sensor accuracy was a function of absolute humidity.*

The IAGOS accuracy is indeed a function of the relative humidity ([https://www.iagos.org/iagos-core-instruments/h2o/](https://www.iagos.org/iagos-core-instruments/h2o/))

*- L126: How does this compare to Wang 2025?*

To further compare with Wang et al. 2025, we have included the following sentence:
    "This differs from Wang et al. (2025) in two ways: i) we use five days for training (four days in Wang et al. (2025)), ii) an additional day gap is used after the validation or testing day to prevent correlations with the following training period."

*- L156: Is it possible the ANN is overfitting these engineered features? You acknowledge the proper data split, but could you use additional data outside the domain to gain confidence?*

We are confident that our 'by day' data splitting strategy avoids overfitting by preventing temporal leakage between the training and test sets. Our objective is to optimize predictions within the North Atlantic region. Applying the model to other domains requires tuning the ANN to fit the specific dynamics of the given region.

*- L160: This criteria sounds more like "No existing cirrus" rather than "clear sky." Could also look at the IAGOS ice crystal measurements to judge pre-existing cirrus (Petzoldt 2025)*

We agree that "No existing cirrus" is a physically accurate description of the condition but we have chosen to use "clear-sky" to stay consistent with previous work in the field (e.g. Wang et al 2025).

*- L170: (Re)Introduce acronym MAE*

We have reintroduced the acronym MAE.

*- L182: Add citation? Where does this baseline come from?*

Table 3 is now referred at the end of the sentence.

*- L186-188: Its not clear to me why "structured input data" ~ drier regimes. Its more clear to me that "high humidity conditions" ~ complex non-linear dependencies.*

We have restructured the paragraph as suggested:
> " The ANN is well-suited for capturing complex, non-linear dependencies, particularly under high humidity conditions, while XGBoost performs robustly in drier regimes."

*- L221-222: This is first clear explanation of why XGBoost is preferable to ANN for the drier regimes. L230 - L233 is also great. Bring this language up front!*
*- Table 3 is super helpful - It would be helpful to use this language up front when describing the benefits of the hybrid architecture.*

We are grateful to Reviewer #1 for these comments. To address them, we modify the abstract to introduce the following sentence at line 11:
> "This hybrid approach significantly outperforms raw ERA5 data, leveraging the ANN's ability to capture non-linear relationships and the XGBoost's robustness in handling drier conditions."

This additional sentence in the abstract necessarily comes with other minor changes to respect the word number limitation, but these do not change the overall content.

*- L223: Repeats the previous line*

- Line 223: We have deleted the previous line (222): "It effectively corrects low RHi values while maintaining a strong improvement in the ETS".

*- Table 3, Table 4: How do these results compare with Wolf et al 2025 or Platt et al 2024 (quantile mapping)*

We have added a sentence at line 236 which compare our results with the ones using quantile mapping by Wolf et al (2025):
> " In comparison, while QM successfully removes bias from the distribution, it does not improve point-wise predictive accuracy. Wolf et al (2025) explicitly report that after applying QM, the ``RMSE, MAE, and MSE increase unnoticeably". In contrast, our model significantly reduces the MAE from $13.70\%$ to $11.37\%$. "

Sincerely,

Jérémie Juvin-Quarroz, on the behalf of the authors